# Simple, Low-Cost and Long-Lasting Film for Virus Inactivation Using Avian Coronavirus Model as Challenge

**DOI:** 10.3390/ijerph17186456

**Published:** 2020-09-04

**Authors:** Luiz Alberto Colnago, Iara Maria Trevisol, Daiane Voss Rech, Lucimara Aparecida Forato, Cirlei Igreja do Nascimento Mitre, José Paulo Gagliardi Leite, Rodrigo Giglioti, Cintia Hiromi Okino

**Affiliations:** 1Embrapa Instrumentação, Rua XV de Novembro 1452, São Carlos 13560-970, Brazil; lucimara.forato@embrapa.br; 2Embrapa Suínos e Aves, Rodovia BR-153 Km 110, Concórdia 89715-899, Brazil; iara.trevisol@embrapa.br (I.M.T.); daiane.rech@embrapa.br (D.V.R.); 3Instituto de Química de São Carlos Universidade de São Paulo, Av. Trabalhador São-Carlense 400, São Carlos 13566-590, Brazil; cirlei.nascimento@hotmail.com; 4Laboratório de Virologia Comparada e Ambiental, Instituto Oswaldo Cruz, Fiocruz, Avenida Brasil, 4365, Rio de Janeiro 21040-360, Brazil; jpgleite@ioc.fiocruz.br; 5Instituto de Zootecnia-IZ, Centro de Pesquisa de Genética e Reprodução Animal, Rua Heitor Penteado n.56, Nova Odessa 13380-011, Brazil; rodrigo.giglioti@sp.gov.br; 6Embrapa Pecuária Sudeste, Rodovia Washington Luiz Km 234 s/n, São Carlos 13560-970, Brazil

**Keywords:** coronavirus, film, detergent, antiviral, virucide, inactivation, sanitization

## Abstract

The COVID-19 infection, caused by SARS-CoV-2, is inequitably distributed and more lethal among populations with lower socioeconomic status. Direct contact with contaminated surfaces has been among the virus sources, as it remains infective up to days. Several disinfectants have been shown to inactivate SARS-CoV-2, but they rapidly evaporate, are flammable or toxic and may be scarce or inexistent for vulnerable populations. Therefore, we are proposing simple, easy to prepare, low-cost and efficient antiviral films, made with a widely available dishwashing detergent, which can be spread on hands and inanimate surfaces and is expected to maintain virucidal activity for longer periods than the current sanitizers. Avian coronavirus (ACoV) was used as model of the challenge to test the antivirus efficacy of the proposed films. Polystyrene petri dishes were covered with a thin layer of detergent formula. After drying, the films were exposed to different virus doses for 10 min and virus infectivity was determined using embryonated chicken eggs, and RNA virus quantification in allantoic fluids by RT-qPCR. The films inactivated the ACoV (ranging from 10^3.7^ to 10^6.7^ EID_50_), which is chemically and morphologically similar to SARS-CoV-2, and may constitute an excellent alternative to minimize the spread of COVID-19.

## 1. Introduction

The World Health Organization (WHO) has declared that COVID-19 is a pandemic, caused by the new human coronavirus (SARS-CoV-2) that has already infected more than 20 million people and cause more than 700 thousand deaths worldwide by August 2020 [1]. Person-to-person contact and contact with contaminated surfaces have been identified as the most common modes of transmission [2]. The length of infectivity time of SARS-CoV-2 was recently evaluated on several inanimate surfaces, and it ranges from hours to several days [3]. The efficiency of surface disinfectant has been investigated and it has been shown to inactivate SARS-CoV-2 in a few minutes [3,4]. However, these virus inactivation agents rapidly evaporate, are flammable, become inefficient soon after application and, consequently, the sanitized surfaces can become a new transmission source following novel contamination. These products are also hard to find or expensive in underdeveloped countries, which are the new hotspots for the disease. Methods for prevention of transmission by contact, through washing hands with soaps and sanitization with alcohols, have been implemented in several countries [5]. However, these procedures are only efficient at eliminating contamination that happened before the sanitization and the hands might be contaminated shortly after these sanitization procedures. Therefore, there is an urgent need for a simple, low-cost and efficient antiviral procedure that is able to maintain virus inactivation efficiency up to several hours.

Our hypothesis was that the use of a long-lasting dry film, made with dishwashing detergent, has the ability to reduce viral infectivity by several orders of magnitude in few minutes on inanimate surfaces. A similar film, applied on hands, could also reduce the risk of contamination when hand washing and/or sanitization are reduced, especially in locations where water is not easily available, which is a common scenario in developing countries where the majority of the population has low socioeconomic status.

Detergents have a high concentration of surfactants such as sodium dodecyl sulfate (SDS), also known as sodium lauryl ether sulfate (SLES) or sodium lauryl sulfate (SLS), which is a well-known protein-denaturing agent [6], and linear alkylbenzene sulfonates (LAS). Therefore, surfactants could denature or induce small conformational changes in the supramolecular structure of S (spike protein) of SARS-CoV-2 that binds to the ACE-2 (angiotensin-converting enzyme 2) receptor of the epithelial cells in the host [2], reducing the virus infectivity. Surfactants may also inactivate enveloped viruses (like SARS-CoV-2), acting on the lipidic layer [7]. Scientific reviews by the independent Cosmetic Ingredient Review (CIR) initiative concluded that SDS is safe and it is not a cause for concern to the consumer [8], the same conclusion was confirmed by another study [9]. In order to verify the potential inactivation efficiency of the detergent dry film on SARS-CoV-2, we have used the avian coronavirus (ACoV), also known as avian infectious bronchitis virus, as model of this challenge, since the chemistry and structure are very similar between coronaviruses [10]. Both viruses are enveloped, single-strand, positive-sense RNA viruses. ACoV belongs to the Gammacoronavirus genus, while SARS-CoV-2 is classified in the Betacoronavirus genus [11]. ACoV was the first coronavirus to be reported in 1937, from chickens with respiratory disease [12]. Due to the huge economic impact caused by this virus, ACoV is one of the most studied coronaviruses in recent decades [13,14]. Among the main advantages of using ACoV as a model of this challenge is the non-zoonotic nature of this virus (the infectivity is restricted to chickens), which can be cultivated under lower biosafety levels. Additionally, there are numerous attenuated virus vaccines that are commercially available, allowing the experiments to be reproduced worldwide. Therefore, this study aimed to evaluate the chemical stability and the ability to inactivate ACoV infectivity of two proposed detergent based films, one for applying to inanimate surfaces and one to be used on hands.

## 2. Materials and Methods

### 2.1. Protocols to Prepare the Antiviral Films

Household dishwashing detergent, Ype Clear (Quimica Amparo LTDA, Amparo, SP, Brazil), contains approximately 8% of surfactant (2% of SDS and 6% of linear alkylbene sulfonates) and other non-surfactants compounds. The soybean oil used was from Soya brand (Bunge, SP, Brazil). Two antiviral films, for application to inanimate surfaces and hands, were evaluated. The film for inanimate surfaces was prepared by diluting the detergent in distilled water (2:1 ratio). The film for hand application was prepared using detergent and soybean oil (20:1 ratio), and the mixture was completely homogenized until a white emulsion was obtained (if stored, the formula must be re-homogenized before use). The oil was added into this last formula as a plasticizer, to improve flexibility [15] and reduce skin drying. Both formulas were immediately used after preparation. Next, 200 µL of each formula was applied to sterile plastic petri dish (85 mm diameter) and spread in its surface to form a thin film. After approximately 30 min the water evaporated and a thin, sticky and translucent dry film was obtained.

### 2.2. Chemical Stability of the Film

The chemical stability of the films was analyzed by high resolution NMR spectroscopy. The ^1^H spectra were acquired on a high-resolution 600 MHz (14.1 T) Avance III NMR spectrometer (Bruker, Karlsruhe, Germany) using a 5 mm broadband probe. The dry films were prepared with 200 µL of the detergent casted on petri dishes. The fresh film was analyzed 30 min after preparation and a second film evaluation was analyzed after 7 days in the laboratory at room temperature. The film components were extracted into 1 mL of deuterated water (D_2_O) and transferred to a 5 mm NMR tube. The spectra were acquired with and without water suppression sequences, using DSS as chemical shift standard. The experiment was performed with 30 ∗ pulses, 3.89 s acquisition time, 64 K data points, spectral width of 14 ppm, 32 scans and recycle delay of 5 s.

### 2.3. Avian Coronavirus (ACoV) as Model for this Challenge

The commercial attenuated ACoV vaccine, H120 strain/BRMSA 1775 from CMISEA (Collection of Microorganisms Important to Swine and Poultry), was used as the challenge model for the antiviral activity of tested films. Three bottles of lyophilized vaccine (1000 doses for each bottle) were diluted into 12 mL of Phosphate-buffered saline (PBS). Titration was carried out by inoculation of five 10-day-old specific pathogen-free (SPF) embryonated chicken eggs per dilution (from a ten-fold serial dilution of the virus) via the allantoic cavity route [16], and the virus titre was expressed as 50% embryo infectious doses (EID_50_/mL) according to Reed and Muench (1938) [17]. The final titre was estimated as 10^7.7^ EID_50_/mL.

### 2.4. Evaluation of the Film’s Antiviral Activity

For each tested film, seven polystyrene petri dishes of 85 mm diameter were used and labelled into A to G (Table 1). Then, 200 µL of film was dispensed in the center of the petri dish from groups A, B and C. The film was uniformly distributed across the whole petri dish surface using a sterile cell scraper. Plates were allowed to dry at room temperature (20 to 25 °C) during 30 to 40 min. The virus levels of 10^6.7^ EID_50_ (high challenge), 10^4.7^ EID_50_ (intermediate challenge) and 10^3.7^ EID_50_ (low challenge) were distributed in groups A/D, B/E and C/F, respectively. Then, 200 µL of virus diluted in PBS pH 7.2 was dispensed per petri dish and incubated for 10 min at room temperature. The virus solution was recovered from each petri dish by washing with 1.8 mL of transport media containing antibiotics (10,000 IU/mL of penicillin G, 5 mg/mL of streptomycin and 0.65 mg/mL of kanamycin sulfate). For each petri dish, 0.2 mL of the recovered suspension was inoculated into the allantoic cavity of 11-day-old SPF embryonated chicken eggs, four to six eggs were inoculated per petri dish. Three additional controls were included (G to I), G was inoculated with PBS recovered from the petri dish containing film but no virus, aiming to confirm absence of toxic effects in the chicken embryo, H was inoculated with transport media and I remained non-inoculated. The inoculated eggs were incubated at 37 °C and candled daily for 7 days, wherein mortality observed in the first 24 h was considered as nonspecific. Typical embryonic changes, consisting of stunted and curled embryos with feather dystrophy (clubbing), were classified as positive for ACoV [16,18]. Allantoic fluid was individually harvested (on the day of embryo death or from all surviving embryos at the seventh day post-inoculation) and stored at −80 °C until it was processed.

### 2.5. Absolute Quantification of Avian Coronavirus (ACoV) RNA Copies by RT-qPCR

RNA extractions from 100 µL of allantoic fluid were performed using TRIzol Reagent (Invitrogen, Carlsbad, CA, USA), followed by RNA purification using an Rneasy Mini Kit (Qiagen^®,^ Hilden, Germany). All samples were tested for ACoV viral load by RT-qPCR (hydrolysis probe system) using an AgPath-ID One step RT-PCR kit (Ambion^®,^ Austin, TX, USA), primers and LNA-probe (5′FAM-3′BHQ1, IDT) for amplification of the 3′UTR genome region of ACoV, as described [19]. The standard curve was constructed using reverse transcribed RNA for estimation of absolute quantification of ACoV RNA copies. Briefly, the RNA extracted from ACov was submitted to conventional RT-PCR as described [19], targeting a fragment of the 3′ UTR of ACoV, 276 nucleotides in size. The PCR product was cloned into a TOPO TA vector (Invitrogen), according to manufacturer instructions and the transformed plasmid was inserted into DH5alpha competent cells. The extracted plasmidial DNA was reverse transcribed using a MEGAscript™(Thermo Fischer Scientific, Waltham, MA, USA) T7 Transcription Kit (Ambion) as recommended by the manufacturer, the transcribed RNA was quantified using a Qubit™ RNA HS Assay Kit (Invitrogen, Carlsbad, CA, USA) and stored at −70 °C until be processed. The estimation of the number of RNA copies was calculated using the formula: {[(g/µL of RNA)/(size of transcribed RNA × 320)]/(6.022 × 10^23^)}. Cq (Cycle quantification) results were used to calculate the number of RNA copies (Log10) using the linear equation from the standard curve. Samples presenting Cq ≤ 36 were classified as positive for ACoV.

### 2.6. Ethical Standards

All the chicken embryos used in this study were handled in accordance to the standards of animal welfare and ethics adopted by the Brazilian Board for Animal Experimentation (Colégio Brasileiro de Experimentação Animal, COBEA), and the experimental animal protocol was approved by the Embrapa Swine and Poultry Ethics Committee for Animal Experimentation (no. 21/2020).

### 2.7. Statistical Analysis

The results of Log_10_ ACoV RNA copies were submitted to a test of normality, and to a Kruskal–Wallis comparison test between treatments previously exposed to the film and the respective positive control group. The qualitative results obtained by virus isolation and RT-qPCR were submitted to a Fisher exact test. All analyses were conducted using SAS^®^ (SAS Institute Inc. 2002, Cary, NC, USA) version 8, and the probability level for significance was set at *p* ≤ 0.05.

## 3. Results 

### 3.1. Evaluation of the Film’s Chemical Stability

The ^1^H NMR spectra of the film extracts showed that the major components of the detergent were SDS and LAS. The spectra presented the two aromatic signals of LAS, between 7 and 8 ppm, some weak peaks between 2 and 4.5 ppm, related to the hydrogen bonded to carbon 2 of SDS, and the other non-surfactant compounds present in the detergent. The strongest NMR signals were observed between 0.5 to 1.5 ppm, which are related to the CH_2_ and CH_3_ groups of SDS and LAS. The spectrum profile obtained from the fresh film was identical to the one obtained after 7 days, with an absence of any new peaks, indicating that during these 7 days, the detergent components remained stable. The long-term chemical stability of the detergent/oil formula was not verified, since it is prepared to be used in the few hours following its creation.

### 3.2. Evaluation of the Film’s Antiviral Activity

The infectivity of the virus suspensions, treated or not treated with tested films, was determined by virus isolation in embryonated chicken eggs and ACoV RNA quantification by RT-qPCR in the allantoic fluid collected from these eggs (Figure 1 and Table 2). The virus suspensions previously exposed to the dry detergent (inanimate surface) and detergent plus vegetable oil (hands) films (groups A, B and C) presented no virus infectivity, as no typical ACoV embryo lesions or positive results by RT-qPCR in the respective allantoic fluids were observed in these groups. On the other hand, all the control virus suspensions (not previously exposed to the films) (groups D, E and F) presented virus infectivity, since typical ACoV lesions in the embryos were induced in all samples and they also presented ACoV RNA positive results—except for one sample from group E (control group of intermediate challenge) which was negative for virus isolation, but was positive for RT-qPCR. The allantoic fluids from eggs inoculated with virus diluent also presented absence of virus activity, as no embryo lesions or positive ACoV results were observed. Significant differences (*p* < 0.05) were observed between treatments previously exposed to film and respective control groups for Log_10_ ACoV RNA copies values and also for qualitative results obtained by virus isolation and RT-qPCR (Appendix A).

## 4. Discussion

In view of the current arising worldwide pandemic distribution of COVID-19, there is a realization that mortality of SARS-CoV-2 is inequitably distributed among vulnerable populations, especially related to lower socioeconomic status. Unlike the high-income countries of Europe, Northern Asia and North America, most countries of lower economic status face limited mitigation capacity, poor access to high quality public health and medical care, dense population and also have scarce access to commercial sanitizers and running water. Alternative formulations, such as the here proposed film for virus inactivation, characterized by easy access, low cost and simple preparation, may constitute a powerful and important tool for COVID-19 prophylaxis in these vulnerable populations.

In the present study, the antiviral activity of a simple and low-cost semi-permanent film was tested on a plastic surface, using ACoV as model. A recent review about the persistence of animal and human coronaviruses on different types of inanimate surfaces showed that coronaviruses remain infectious on a plastic surface for the same time or longer than other surfaces (steel, aluminum, wood, paper, glass, silicon rubber, latex, disposable gown, ceramic and teflon) [20,21]. The longest viral viability time (9 days) was found on a plastic surface [21]. Together, these results indicate that hard and non-porous plastic surfaces, as used in this study, are a good choice to screen the inactivation effects of disinfectants. Equally, despite the fact that this study has used ACoV as a virus model, the obtained findings may be extended to SARS-CoV-2, as similar chemical composition and structure were verified for the viruses of the *Coronaviridae* family [10]. Similar environmental resistance was observed between these two viruses. At 56 °C, SARS-CoV-2 was viable after 10 min and inactivated after 30 min [3], while nine different ACoV strains were inactivated after 15 min at the same temperature [13]. Both, SARS-CoV-2 and ACoV were extremely stable in a wide range of pH values (3–10) at room temperature [3,13].

The presence of SARS-CoV-2 on surfaces is always a concern. In this study, film on a plastic surface was able to inactivate the ACoV at all tested challenge doses. As result, no RNA detection (RT-qPCR) or virus activity in chicken embryos was observed in treatments previously exposed to the film (Figure 1 and Table 2), while all positive controls (challenged and not previously exposed to the film) presented virus replication observed by macroscopic lesions and RNA virus detection in the embryos. 

The antiviral activity of the film can be attributed, in large part, to the biocidal action of the surfactants present in the detergent. Surfactants (surface active agents) are the single most important ingredients in laundry and household cleaning products [9,22], comprising 1% to 30% in cleaning products formulation [9]. SLS surfactant could be a potent inhibitor of the infectivity of different types of pathogens without causing marked toxicity to skin and/or mucosae [23]. The mechanism by which SLS inactivates enveloped and non-enveloped viruses probably involves the denaturation of envelope or capsid proteins. These proteins may play different roles in the viral replicative cycle such as adhesion receptors, proteins involved in the encapsulation of viral genome [23]. Protein denaturation involves an initial rapid process where protein and SLS produce aggregates, followed by two slower processes, where the complexes first disaggregate into single protein structures situated asymmetrically on the SLS micelles, followed by the isotropic redistribution of the protein [6]. 

Although the viral load of coronaviruses on inanimate surfaces is not clearly known to contribute as a source of contamination, during an outbreak, it seems plausible that reduction in the viral load on surfaces by disinfection, especially those frequently touched by infected patients, may constitute an efficient tool for minimizing the virus spread [20]. The prevalence of face-touching behavior in students was determined as, on average, 23 times per hour, whereas for mucosae (eye, mouth and nose) touching it was 10 times per hour [24]. Consequently, sanitization of both surfaces and hands are essential and inexpensive preventive methods for breaking the transmission cycle. Several biocidal agents, such as 0.5% hydrogen peroxide, 62–71% ethanol or 0.1% of sodium hypochlorite, were able to inactivate coronavirus from inanimate surfaces within one minute [20]. However, these biocidal agents rapidly evaporate or become inefficient in a short period and, consequently, these sanitized surfaces can become a new transmission source following novel contamination.

The results of the chemical stability evaluation of the films here tested showed that SDS and LAS compounds are highly stable in dry film casted on a plastic surface for at least seven days. The chemical stability of the proposed films indicates that the compounds properties, including antiviral activity, will be preserved for the same period, and may keep a residual protective effect. Our proposal is that the detergent film, applied on plastic, metal, glass, ceramic, laminated and other inanimate surfaces, in public areas, can be an efficient alternative to prevent SARS-CoV-2 spread, especially in locations where methods of prophylaxis are scarce. Besides, diluted hand soap (1:49), which is well known to contain surfactants in the composition, was able to reduce 3.6 Log_10_ SARS-CoV-2 after 5 min and reached undetected levels after 15 min [3], reinforcing the role of these substances in virus inactivation.

Therefore, the dry film might effectively maintain virus inactivation ability through hours to days, reducing the use of hazardous chemicals and the need for frequent sanitization procedures with the chemical compounds normally dissolved in water, which generate large amounts of toxic waste that may contaminate ecosystems. New detergent film can be applied to inanimate surfaces without removing the previous detergents, this reduces the need for water and consequently reduces environmental contamination. Hand lotion prepared with detergent/vegetable oil could also reduce the risk of contamination, especially in locations where water is not easily available for hand washing, which is a common scenario in populations in underdeveloped countries.

## 5. Conclusions

The antiviral film we tested demonstrated excellent chemical stability and ability to inactivate ACoV infection at the three challenge doses tested. Further analysis will be performed regarding the shelf life of the proposed films on different inanimate materials, on hand application and with different virus models of challenge. Therefore, this study has demonstrated the efficacy of virus inactivation of a simple, low-cost and easy to prepare long-lasting detergent film, which may constitute an excellent alternative to mitigate the spread of SARS-CoV-2, particularly in populations of low-income countries, as well as opening perspectives for studies with other enveloped viruses.

## Figures and Tables

**Figure 1 ijerph-17-06456-f001:**
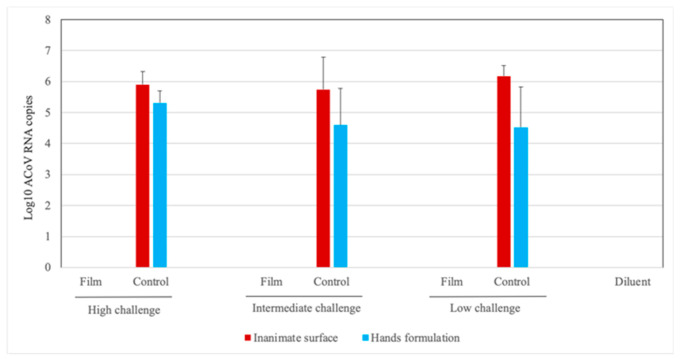
Absolute quantification of Log_10_ avian coronavirus (ACoV) RNA copies (only positive samples presenting Cycle quantification (Cq) values < 36 were included) in the allantoic fluid from embryonated chicken eggs inoculated with virus suspension from different experimental groups previously treated or not treated with film and challenged with high, intermediate and low doses of ACoV.

**Table 1 ijerph-17-06456-t001:** Experimental design used for evaluation of the film’s antiviral activity for hands formulation and inanimate surface film.

Treatment	Description	Film	ACoV Level
A–Film + High challenge	Efficacy of film	Yes	10^6.7^ EID_50_
B–Film + Intermediate challenge	Efficacy of film	Yes	10^4.7^ EID_50_
C–Film + Low challenge	Efficacy of film	Yes	10^3.7^ EID_50_
D–High challenge	Positive control	No	10^6.7^ EID_50_
E–Intermediate challenge	Positive control	No	10^4.7^ EID_50_
F–Low challenge	Positive control	No	10^3.7^ EID_50_
G–Film + Transport medium	Safety control	Yes	No
H–Transport medium	Negative control	No	No
I–Non- inoculated	Embryo control	No	No

**Table 2 ijerph-17-06456-t002:** Qualitative results of embryonated chicken eggs from different experimental groups obtained from avian coronavirus (ACoV) isolation and RT-qPCR.

Experimental Treatments		Positive/Total Samples (% Positive)			
		Virus Isolation ^1^		RT-qPCR ^2^	
		Hands Formulation	Inanimate Surface	Hands Formulation	Inanimate Surface
High Challenge	A–Film	0/5 ^3^ (0%)	0/6 (0%)	0/5 ^3^ (0%)	0/6 (0%)
	D–Control	6/6 (100%)	5/5 (100%)	6/6 (100%)	5/5 (100%)
Intermediate Challenge	B–Film	0/6 (0%)	0/6 (0%)	0/6 (0%)	0/6 (0%)
	E–Control	5/5 ^3^ (100%)	4/5 (80%)	5/5 ^3^ (100%)	5/5 (100%)
Low Challenge	C–Film	0/6 (0%)	0/6 (0%)	0/6 (0%)	0/6 (0%)
	F–Control	6/6 (100%)	5/5 (100%)	6/6 (100%)	5/5 (100%)
	G–Film control	0/6 (0%)	0/4 (0%)	0/6 (0%)	0/4 (0%)
	H-Negative control	0/6 (0%)	0/6 (0%)	0/6 (0%)	0/6 (0%)

^1^ Embryos presenting stunting, curling and/or feather dystrophy (clubbing) were considered as positive for virus isolation. ^2^ Samples presenting Cq values < 36 were considered as positive. ^3^ There was unspecific mortality of one embryo at 24 hours post-infection (hpi), therefore this embryo was not considered in the analysis.

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
