# Peer review of "Simple, Low-Cost and Long-Lasting Film for Virus Inactivation Using Avian Coronavirus Model as Challenge"

_ijerph, 2020, doi:10.3390/ijerph17186456_

Round 1

Reviewer 1 Report

The topic of the manuscript is not without merit and it should be published. It requires only minor revision focused on statistical methods.

Table 1

It would be suitable displayed also number of replicates.

2.7. Statistical analysis

Tukey test is as part of one- and two-way ANOVA. It should be described if quantitative data passed the test of normality, which of test was used etc. 

It should be reported what quantitative variables were statistically tested. 

In addition, sample size seems to be very small (5-6 replicates). 

In chapter 3.2., there are reported: "Significant differences (P < 0.05), were observed between groups previously exposed to film and control group". The p-value could not be calculated by ANOVA with Tukey comparison test because Table 2 documented proportions or positivity rate  (i.e. qualitative variable). The comparison between test and control subgroups should be done with Fisher exact test or chi-2 test.

Author Response

Reviewer #1:    

The topic of the manuscript is not without merit and it should be published. It requires only minor revision focused on statistical methods.

Our reply: We thank the reviewer for his/her thoroughness and expertise.  We appreciated also the time dedicated to do the critical review of our manuscript and, especially, for his (/her) constructive suggestions. We have made several modifications in the manuscript according with recommended by the reviewers. Now we will present our considerations to the specific comments of reviewer-1:

Table 1

It would be suitable displayed also number of replicates.

Our reply: We have modified as recommended.

2.7. Statistical analysis

Tukey test is as part of one- and two-way ANOVA. It should be described if quantitative data passed the test of normality, which of test was used etc.

It should be reported what quantitative variables were statistically tested.

In addition, sample size seems to be very small (5-6 replicates).

In chapter 3.2., there are reported: "Significant differences (P < 0.05), were observed between groups previously exposed to film and control group". The p-value could not be calculated by ANOVA with Tukey comparison test because Table 2 documented proportions or positivity rate (i.e. qualitative variable). The comparison between test and control subgroups should be done with Fisher exact test or chi-2 test

Our reply: We have modified as recommended. We performed the test of normality, but as we found not normally distributed data, we have applied non-parametric test (Kruskal-Wallis) comparison test (instead Tukey test) for Log10 ACoV RNA copies. While the fisher exact test was applied for comparison of qualitative results. A supplementary material was included with all p values obtained.

Reviewer 2 Report

The straight-forward experiment reported in this manuscript addresses the need for a long-lasting surface disinfectant active against coronaviruses as an alternative to currently used disinfectants that might not be readily available in developing countries. A thin film of an evidently commercial detergent containing 2% SDS and 6% linear alkylbene sulfomates deposited in the bottom of petri dishes and allowed to dry was tested for virucidal activity using a vaccine strain of the well-studied avian coronavirus infectious bronchitis virus. After incubation of virus suspended in PBS on the film surface for 10 minutes, any remaining infectious virus was detected by inoculation into embryonated chicken eggs. Evidence of virus replication in the embryos, such as death and IBV-typical embryonic lesions or demonstration of virus in the allantoic fluid by detection of viral RNA by RT-qPCR would indicate the presence of infectious virus. However, no infectious virus was detected after incubation on the film-coated surfaces, while virus preparations incubated on surfaces without the film were still infectious. Two formulations were tested, one intended for use on inanimate surfaces, containing just detergent, and one intended for use on hands, which contained soybean oil in addition to detergent. However, both formulations were tested only on the inanimate petri dishes. The authors implied that similar formulations might be achieved using household dishwashing detergents, but did not test any made from household detergents. (I check the ingredients on my household dishwashing detergent and found it did not contain linear alkylbene sulfomates, but did contain SDS.) The work would have been much more compelling if the initial experiment had been followed up with an experiment showing that dishwashing detergent bought in a store (at least two different brands) could be used, instead of just a commercially-obtained detergent.  This work should be considered a pilot or proof-of-concept study. Many unknowns remain. Only virus in PBS was tested.  Would virus in fast-drying respiratory droplets also be inactivated? This would be difficult to test, unless the virus deposited on the plates via droplets could be recovered without adding liquid, which redissolves the detergent. How long does it take for inactivation to occur? Only one length of time (10 minutes) was tested. If students touch mucosa on their face on the average of ten times per hour, as indicated in the discussion (some students more frequently), then 10 minutes might be too long. As already alluded to, would films generated from household dishwashing detergents also be effective? Could the detergent be diluted more, to make the film less sticky? Whether a sticky film is desirable on high touch surfaces such as doorknobs and whether it would be maintained after being touched multiple times were not addressed. Another consideration not mentioned was whether allowing detergent to dry on the hands would irritate the hands.  Determining this would have to be done in a literally blinded study. People that it was being tested on should not be allowed to look and see the suds while it was being applied to hands and drying. Including the oil in a control formulation might help to make the “feel” of it more similar.

Other comments:

  1. The authors describe the antiviral film they generated as “semi-permanent.” Permanent is an absolute. Something is either permanent or not. It cannot be semi-permanent. A better description might be “long-lasting.” Furthermore, the film applied to skin is unlikely to remain for extended periods of time due to friction and sloughing.

  1. The abstract states that their film maintains virucidal activity longer than current sanitizers. However, the authors only tested viricidal activity shortly after depositing the film, and did not compare the length of time it was active with other sanitizers. The sentence should be rewritten to say that their film is expected to maintain virucidal activity longer than current sanitizers.

  1. I suggest that the first sentence in the introduction include a date at end of sentence, e.g. “by July 2020,” since these numbers will be constantly changing. These numbers should be updated with the most recent figures and date available when the manuscript is revised.
  2. Throughout the manuscript there are too many significant digits in the titers reported. In the section on calculation of titers by the Reed and Muench method in the chapter by Villegas in the Laboratory Manual for the Isolation, Identification, and Characterization of Avian Pathogens, it states: “It is usual to include in the expression of log titers only one figure to the right of the decimal point (10^5.8 LD50/ml, not 10^5.8433). Extension of additional figures in the mantissa implies a degree of accuracy that is not real.” Thus virus amounts should be 10^3.7 – 10^6.7 instead of 10^3.66-10^6.66.
  3. Inconsistent name of virus: SARS-CoV-2, SARSCoV-2, SARS-CoV2 are all used in the mansucript.
  4. There are aspects of Materials & Methods that are unclear or inaccurate:
    1. Coffee spoon and dessert spoon are not concepts in every country.
    2. How was the mixture homogenized? Was a method used that is available in every household, such as vigorous shaking? Does it need to be re-homogenized before every use? (It is not practical to make it immediately before every use.)
    3. Lines 97-98 are unclear. What size petri dish? Was 200 ml/dish used? Was that left to dry or poured off so only a thin film remained? Based on line 113, the authors likely mean 200 µl instead of 200 ml. The procedure is adequately described in lines 113-116, so that section could be referred to here.
    4. Line 107 is unclear. Was each bottle 1000 doses, or together the three bottles 1000 doses? (likely the former)
    5. Lines 108-109 are likely not an accurate description. The authors likely inoculated five eggs with each dilution of a ten-fold serial dilution series.
    6. Line 117 and Table 1. Are the titers given the EID50/ml or the amount of the virus applied to the plate (the amount of virus in 200 µl)? If the latter, then it is not a titer, but just a virus level (without a volume unit).
    7. Line 123. “six eggs were inoculated per group” This is not clear. Each group had six plates. Was one egg/plate inoculated? (I hope the virus from the six plates was not pooled and then inoculated into six eggs. If so, the N of the experiment is 1.)
    8. Line 124 states that the G group of eggs was inoculated with film. It was actually inoculated with PBS recovered from plates with film but no virus.
    9. Line 129: At what time point was allantoic fluid collected? (I assume it was collected from eggs with dead embryos on the day of their death and from all surviving embryos 7 days post-inoculation.)
    10. Line 139. The standard curve is not described. How was it generated? Using in vitro-transcribed RNA, viral RNA from highly purified virions?
    11. Regarding statistical analysis, the description “quantitative parameters” is vague. The only data for which the Tukey test would be valid would be the RT-qPCR results, so it should be stated that the Tukey test was used for the RT-qPCR results. The positive vs. negative virus isolation results could also be analyzed statistically, by Chi squared or Fisher’s exact test
  5. Lines 163-164 indicate that there were almost 3,000-fold fewer viral RNA copies than EID50 in the inocula. The authors likely have not adjusted the virus levels detected in the volume of RNA used in the RT-qPCR to the number of copies that would be present in the inocula.
  6. Table 1:
    1. Center alignment of the first column makes the table difficult to read. Also, I suggest using two columns for this, the first column being Group and the second column being Description.
    2. The group descriptions are not consistently formatted. Groups D-F indicate that they are controls in parentheses, whereas group H has control in the main description of the group. For consistency, group H description should be “Transport medium (negative control)”
    3. The last column should have titer (or titre) rather than titter in the heading.
  7. Table 2: A footnote indicates that group A had an unspecific mortality, resulting in only 5 embryos analyzed. In group G inanimate surface there are only 4 embryos. Was there also non-specific mortality in this group, indicating possible toxicity of the formulation?
  8. Headings in Table 2 and key in Fig. 1 are misleading. No testing on hands was conducted. “Hands formulation,” might be a more accurate label. Even “hands film” as used in the text (line 170) is misleading because the film was not tested on hands.
  9. Fig.1 is lacking error bars, for the results from the six plates (N=6).
  10. Regarding references cited. If reference 21 showed the longest viability time on plastic, then both references 20 and 21 should be cited for the previous sentence.

Language corrections that affect meaning:

  1. Lines 28, 113, 114, 115, 119, 120: 90 mm plates are not microplates. Do the authors wish to convey the information that these are plates normally for microbiology in contrast to plates normally used for eukaryotic cell culture? If so they can call them petri dishes, as they have in line 98
  2. Line 32: A comma is needed after SARS-CoV-2. As written the sentence says that ACoV may constitute an excellent alternative to minimize the spread of Covid-19
  3. Lines 40-41 say that infectivity ranges from hours to several days. Hours and days are not units for infectivity. Instead of “infectivity,” one correct possibility would be “The length of time infectivity is maintained.
  4. Line 75: Although “innumerous” means “innumerable,” “innumerable” is more commonly used. However, either is any overstatement. Saying there are numerous attenuated vaccines commercially available would be more accurate.
  5. I think the authors mean dishwashing detergent instead of dishwasher detergent. At least in the United States, dishwasher detergent is a powder or cake used in a dishwashing machine, whereas dishwashing detergent is a liquid concentrate used when washing dishes in a sink.
  6.  

Other types of language errors:

Adjective used instead of adverb and vice versa

Adjective used instead of noun

Verb form errors: Subject-verb agreement, using conjugated verb where infinitive should be used, using infinitive when gerund should be used.

A . . . then plural (e.g. lines 24-25)

Awkward word choices (e.g. in line 40 it should say “identified” instead of “pointed.” If the authors want to include “pointed,” it should say “pointed out” or “pinpointed.”

Wrong preposition: e.g. line 41. It should be “on surfaces” instead of “in surfaces.” This error (using “in” when “on” should be used) is present many times in the manuscript.

Punctuation (e.g. using comma where semicolon should be used in compound sentence)

Using s on adjectives for plural nouns. Singular form should always be used when a noun is used as an adjective.

Lack of space between number and units. (It should be 100 µl instead of 100µl.)

Unnecessary “the” (e.g. line 152 “the SDS”)

Incomplete sentence (lines 206-206)

Misspelled word (forms in line 231)

Author Response

Reviewer #2: The straight-forward experiment reported in this manuscript addresses the
need for a long-lasting surface disinfectant active against coronaviruses as an alternative
to currently used disinfectants that might not be readily available in developing
countries. A thin film of an evidently commercial detergent containing 2% SDS and 6%
linear alkylbene sulfomates deposited in the bottom of petri dishes and allowed to dry
was tested for virucidal activity using a vaccine strain of the well-studied avian
coronavirus infectious bronchitis virus. After incubation of virus suspended in PBS on
the film surface for 10 minutes, any remaining infectious virus was detected by
inoculation into embryonated chicken eggs. Evidence of virus replication in the
embryos, such as death and IBV-typical embryonic lesions or demonstration of virus in
the allantoic fluid by detection of viral RNA by RT-qPCR would indicate the presence
of infectious virus. However, no infectious virus was detected after incubation on the
film-coated surfaces, while virus preparations incubated on surfaces without the film
were still infectious. Two formulations were tested, one intended for use on inanimate
surfaces, containing just detergent, and one intended for use on hands, which contained
soybean oil in addition to detergent.
Our reply: We thank the reviewer for his/her thoroughness and expertise. We
appreciated also the time dedicated to do the critical review of our manuscript and,
especially, for his (/her) constructive suggestions. We have made several modifications
in the manuscript according with recommended by the reviewers. Now we will present
our considerations to the specific comments of reviewer-2:
However, both formulations were tested only on the inanimate petri dishes. The authors
implied that similar formulations might be achieved using household dishwashing
detergents, but did not test any made from household detergents. (I check the
ingredients on my household dishwashing detergent and found it did not contain linear
alkylbene sulfomates, but did contain SDS.)
Our reply: We would like to better clarify our method. The dishwashing detergent was
a household detergent commonly found in Brazilian markets. We have double checked
the composition of three different brands: Ype (used in our experiment), Limpol and
triex, and all three household detergents contain linear alkylbene sulfonates as main
component. See the pictures below:
From the left to the right: Triex, Limpol and Ype brands of dishwashing detergent.
The work would have been much more compelling if the initial experiment had been
followed up with an experiment showing that dishwashing detergent bought in a store
(at least two different brands) could be used, instead of just a commercially-obtained
detergent.
Our reply: As we mentioned before, our commercial household dishwashing detergents
were composed by same active compound: linear alkylbene sulfonates. Then, we
hypothesized that testing one of those brands could be representative for most
dishwashing detergents.
This work should be considered a pilot or proof-of-concept study. Many unknowns
remain. Only virus in PBS was tested. Would virus in fast-drying respiratory droplets
also be inactivated? This would be difficult to test, unless the virus deposited on the
plates via droplets could be recovered without adding liquid, which redissolves the
detergent. How long does it take for inactivation to occur? Only one length of time (10
minutes) was tested.
Our reply: We are in agreement with the reviewer’s statement, this study is considered
a pilot study. Further experiments are already being performed, to better elucidate these
questions. We hope that in a short future we could also present these answers.
If students touch mucosa on their face on the average of ten times per hour, as indicated
in the discussion (some students more frequently), then 10 minutes might be too long.
Our reply: We are partially in agreement with the reviewer’s statement. As we
mentioned before, further experiments are being performed, including testing other
intervals of time. Furthermore, as we just tested one interval (10 minutes), shorter
intervals of time will be also tested in order to estimate the time required for complete
virus inactivation. Besides, it’s important to highlight that in just 10 minutes, the film was
able to inactivate up to a million of active virus, therefore we are sure that even if we
confirm that all the 10 minutes are necessary to completely inactivate the virus, our
proposed films have great value in locations where no other prevention tools are
reachable.
As already alluded to, would films generated from household dishwashing detergents also
be effective?
Our reply: We would like to emphasize that the films were prepared with a household
dishwashing detergent (Ype) commercially available in markets.
Could the detergent be diluted more, to make the film less sticky?
Our reply: Yes, we have prepared several films using different detergent dilutions and
vegetable oil concentrations and we are planning to test these other formulations as soon
as possible.
Whether a sticky film is desirable on high touch surfaces such as doorknobs and
whether it would be maintained after being touched multiple times were not addressed.
Our reply: We guess that sticky films are ideal for this application as indicated in our
manuscript. But if the film is being touched multiple times, the surface will need new
detergent applications.
Another consideration not mentioned was whether allowing detergent to dry on the
hands would irritate the hands. Determining this would have to be done in a literally
blinded study. People that it was being tested on should not be allowed to look and see
the suds while it was being applied to hands and drying. Including the oil in a control
formulation might help to make the “feel” of it more similar.
Our reply: We are in agreement with the reviewer about the requirement of a blinded
skin test and we are also planning to perform it as soon as possible. However, we would
like to emphasize is that surfactants like SDS or SLS are widely used in cosmetics
products. In this context, the US Cosmetic Ingredient Review (CIR) and Human and
Environmental Risk Assessment (HERA) initiative concluded that the SDS is safe and
not a cause for concern to consumer use (CRI 2005, Bondi 2015). Indeed, one of the
adverse effects of soaps and detergents is dry skin, but it can be minimizing using
lotion and body cream overnight to help rehydrate the skin (Beldon, 2012).
Besides, since the current pandemic is characterized by the urgency for new
alternative prophylactic solutions, especially in poor communities, we guess that
a blinded study and testing a control formulation with oil without the detergent can
be performed in future experiments.
References:
RI ( Cosmetic Ingredient Review (CIR) Final report on the safety assessment of sodium
lauryl sulfate and ammonium lauryl sulfate. Int J Toxicol. 2005;24(1):1–102).
Bondi 2015 Human and Environmental Toxicity of Sodium Lauryl Sulfate (SLS):
Evidence for Safe Use in Household Cleaning Products
Beldon P (2012) The latest advances in skin protection. Wounds UK 8(2): S17–9
Other comments:
1. The authors describe the antiviral film they generated as “semi-permanent.”
Permanent is an absolute. Something is either permanent or not. It cannot be semipermanent.
A better description might be “long-lasting.” Furthermore, the film applied
to skin is unlikely to remain for extended periods of time due to friction and sloughing.
Our reply: We have modified as recommended. All the “semi-permanent” citations
were replaced by “long-lasting”, including the title.
2. The abstract states that their film maintains virucidal activity longer than current
sanitizers. However, the authors only tested viricidal activity shortly after depositing the
film, and did not compare the length of time it was active with other sanitizers. The
sentence should be rewritten to say that their film is expected to maintain virucidal
activity longer than current sanitizers.
Our reply: We have modified as recommended (P1L29).
3. I suggest that the first sentence in the introduction include a date at end of sentence,
e.g. “by July 2020,” since these numbers will be constantly changing. These numbers
should be updated with the most recent figures and date available when the manuscript
is revised.
Our reply: We have modified as recommended (P1-L42).
4. Throughout the manuscript there are too many significant digits in the titers reported.
In the section on calculation of titers by the Reed and Muench method in the chapter by
Villegas in the Laboratory Manual for the Isolation, Identification, and Characterization
of Avian Pathogens, it states: “It is usual to include in the expression of log titers only
one figure to the right of the decimal point (10^5.8 LD50/ml, not 10^5.8433). Extension
of additional figures in the mantissa implies a degree of accuracy that is not real.” Thus
virus amounts should be 10^3.7 – 10^6.7 instead of 10^3.66-10^6.66.
Our reply: We have modified as recommended.
5. Inconsistent name of virus: SARS-CoV-2, SARSCoV-2, SARS-CoV2 are all used in
the mansucript.
Our reply: We have modified as recommended. We have adopted SARS-CoV-2 as
standard nomenclature.
6. There are aspects of Materials & Methods that are unclear or inaccurate:
a. Coffee spoon and dessert spoon are not concepts in every country.
Our reply: We are in agreement with the reviewer’s statement. We have removed the
spoons as units of measure from the manuscript.
b. How was the mixture homogenized? Was a method used that is available in every
household, such as vigorous shaking? Does it need to be re-homogenized before every
use? (It is not practical to make it immediately before every use.)
Our reply: The mixture was homogenized in vortex shaker, but at home, the mixture
may be vigorous shaking, and must be re-homogenized before every use. We modified
the text as recommended. The sentence “if stored the formula must be re-homogenized
before use” was inserted at P2L91.
c. Lines 97-98 are unclear. What size petri dish? Was 200 ml/dish used? Was that left to
dry or poured off so only a thin film remained? Based on line 113, the authors likely
mean 200 μl instead of 200 ml. The procedure is adequately described in lines 113-116,
so that section could be referred to here.
Our reply: We have used petri dishes with 85 mm of diameter. 200 μL of each formula
was applied per petri dish, and the films were left to dry. We have corrected the
sentence changing mL per μL.
d. Line 107 is unclear. Was each bottle 1000 doses, or together the three bottles 1000
doses? (likely the former)
Our reply: Each bottle contained 1000 doses. We have inserted the sentence “1000
doses for each bottle” at P3L110-111.
e. Lines 108-109 are likely not an accurate description. The authors likely inoculated
five eggs with each dilution of a ten-fold serial dilution series.
Our reply: Four to six eggs were inoculated per dilution, from a ten-fold serial dilution
series. We have modified the text to better clarify (P3L126-127).
f. Line 117 and Table 1. Are the titers given the EID50/ml or the amount of the virus
applied to the plate (the amount of virus in 200 μl)? If the latter, then it is not a titer, but
just a virus level (without a volume unit).
Our reply: The EID50 informed is related to the amount of virus applied to the plate
(the amount of virus in 200 μl. We have modified the text according to the
recommendation.
g. Line 123. “six eggs were inoculated per group” This is not clear. Each group had six
plates. Was one egg/plate inoculated? (I hope the virus from the six plates was not
pooled and then inoculated into six eggs. If so, the N of the experiment is 1.)
Our reply: We have understood the point of view of the reviewer. We have inoculated
six or five eggs per group, then the sampling would be one. However, we must take
some attention to the fact that three different doses were tested for each film. Then, the
detergent action of virus inactivation was observed at 6 different times. We are in
agreement that further experiments must be performed in order to answer all the
questions raised by the reviewer, and as we mentioned before, new studies are already
being carried out. But the present study may be considered the first step, and we guess
that the strength and importance of our results cannot be ignored. Therefore, we would
like to highlight that an earlier disclosure of our results, even if considered preliminary
(a pilot study), may stimulate new groups to better evaluate our inedited findings.
h. Line 124 states that the G group of eggs was inoculated with film. It was actually
inoculated with PBS recovered from plates with film but no virus.
Our reply: We are in agreement with the reviewer’s statement, then we have modified
the sentence for “Three additional groups were included, G group was inoculated with
PBS recovered from the microplate containing film but no virus…” (P3L127-130).
i. Line 129: At what time point was allantoic fluid collected? (I assume it was collected
from eggs with dead embryos on the day of their death and from all surviving embryos
7 days post-inoculation.)
Our reply: Yes, the reviewer is right. We have inserted the information “Allantoic fluid
was individually harvested (on the day of embryo death and from all surviving embryos
at seventh day post-inoculation) and stored at -80°C until be processed.” in the
manuscript (P3-L133-135).
j. Line 139. The standard curve is not described. How was it generated? Using in vitrotranscribed
RNA, viral RNA from highly purified virions?
Our reply: We have used in vitro transcribed RNA. Briefly, the RNA extracted from
Acov was submitted to conventional RT-PCR as described [19], targeting a fragment of
3’ UTR of AcoV, with 276 bp of size. The PCR product was cloned into TOPO TA
vector (Invitrogen), according to manufacturer instructions and the transformed plasmid
was inserted in DH5alpha competent cells. The extracted plasmidial DNA was reverse
transcribed using MEGAscript™ T7 Transcription Kit (Ambion) as recommended by
manufacturer, the transcribed RNA was quantified using Qubit™ RNA HS Assay Kit
(Invitrogen) and stored at -70°C until be processed. The estimation of the number of
RNA copies was calculated using the formula: {[(g/ μL of RNA)/(size of transcribed
RNA * 320)]/(6.022*1023)]. We have inserted this information in the manuscript.
k. Regarding statistical analysis, the description “quantitative parameters” is vague. The
only data for which the Tukey test would be valid would be the RT-qPCR results, so it
should be stated that the Tukey test was used for the RT-qPCR results. The positive vs.
negative virus isolation results could also be analyzed statistically, by Chi squared or
Fisher’s exact test.
Our reply: We have modified as recommended. We performed the test of normality,
but as we found not normally distributed data, we have applied non-parametric test
(Kruskal-Wallis) comparison test (instead Tukey test) for Log10 ACoV RNA copies.
While the fisher exact test was applied for comparison of qualitative results. A
supplementary material was included with all p values obtained.
7. Lines 163-164 indicate that there were almost 3,000-fold fewer viral RNA copies
than EID50 in the inocula. The authors likely have not adjusted the virus levels detected
in the volume of RNA used in the RT-qPCR to the number of copies that would be
present in the inocula.
Our reply: We would like to better clarify about the question raised by the reviewer.
The numeric values obtained from virus titration is not necessarily correlated to the
values obtained from RT-qPCR. Since virus isolation method is related to the dose that
induces lesions in at least 50% of the embryos, while RT-qPCR is related to RNA virus
quantification which may be derived from active or inactivated virus. Therefore, we
cannot point out the reason to adjust the virus levels detected in volume of RNA to the
number of RNA copies present in the inocula.
8. Table 1:
a. Center alignment of the first column makes the table difficult to read. Also, I suggest
using two columns for this, the first column being Group and the second column being
Description.
Our reply: We have modified as recommended.
b. The group descriptions are not consistently formatted. Groups D-F indicate that they
are controls in parentheses, whereas group H has control in the main description of the
group. For consistency, group H description should be “Transport medium (negative
control)”
Our reply: We have modified as recommended.
c. The last column should have titer (or titre) rather than titter in the heading.
Our reply: We have modified as recommended.
9. Table 2: A footnote indicates that group A had an unspecific mortality, resulting in
only 5 embryos analyzed. In group G inanimate surface there are only 4 embryos. Was
there also non-specific mortality in this group, indicating possible toxicity of the
formulation?
Our reply: No, we have no non-specific mortality in group G inanimate surface. Due to
limited number of SPF embryonated chicken eggs, we have to just four eggs in this
experimental group.
10. Headings in Table 2 and key in Fig. 1 are misleading. No testing on hands was
conducted. “Hands formulation,” might be a more accurate label. Even “hands film” as
used in the text (line 170) is misleading because the film was not tested on hands.
Our reply: We have modified as recommended.
11. Fig.1 is lacking error bars, for the results from the six plates (N=6).
Our reply: We have modified as recommended.
12. Regarding references cited. If reference 21 showed the longest viability time on
plastic, then both references 20 and 21 should be cited for the previous sentence.
Our reply: We have modified as recommended.
Language corrections that affect meaning:
1. Lines 28, 113, 114, 115, 119, 120: 90 mm plates are not microplates. Do the authors
wish to convey the information that these are plates normally for microbiology in
contrast to plates normally used for eukaryotic cell culture? If so they can call them
petri dishes, as they have in line 98
Our reply: We have modified as recommended.
2. Line 32: A comma is needed after SARS-CoV-2. As written the sentence says that
ACoV may constitute an excellent alternative to minimize the spread of Covid-19
Our reply: We have modified as recommended.
3. Lines 40-41 say that infectivity ranges from hours to several days. Hours and days are
not units for infectivity. Instead of “infectivity,” one correct possibility would be “The
length of time infectivity is maintained.
Our reply: We have modified as recommended.
4. Line 75: Although “innumerous” means “innumerable,” “innumerable” is more
commonly used. However, either is any overstatement. Saying there are numerous
attenuated vaccines commercially available would be more accurate.
Our reply: We have modified as recommended.
5. I think the authors mean dishwashing detergent instead of dishwasher detergent. At
least in the United States, dishwasher detergent is a powder or cake used in a
dishwashing machine, whereas dishwashing detergent is a liquid concentrate used when
washing dishes in a sink.
Our reply: We have modified as recommended.
Other types of language errors:
Adjective used instead of adverb and vice versa
Adjective used instead of noun
Verb form errors: Subject-verb agreement, using conjugated verb where infinitive
should be used, using infinitive when gerund should be used.
Our reply: We have carefully revised the whole manuscript aiming to correct the cited
language errors.
A . . . then plural (e.g. lines 24-25)
Our reply: We have modified as recommended.
Awkward word choices (e.g. in line 40 it should say “identified” instead of “pointed.” If
the authors want to include “pointed,” it should say “pointed out” or “pinpointed.”
Our reply: We have modified as recommended.
Wrong preposition: e.g. line 41. It should be “on surfaces” instead of “in surfaces.” This
error (using “in” when “on” should be used) is present many times in the manuscript.
Our reply: We have modified as recommended.
Punctuation (e.g. using comma where semicolon should be used in compound sentence)
Our reply: We have carefully revised the whole manuscript aiming to correct the cited
language errors.
Using s on adjectives for plural nouns. Singular form should always be used when a
noun is used as an adjective.
Our reply: We have carefully revised the whole manuscript aiming to correct the cited
language errors.
Lack of space between number and units. (It should be 100 μl instead of 100μl.)
Our reply: We have modified as recommended.
Unnecessary “the” (e.g. line 152 “the SDS”)
Our reply: We have modified as recommended.
Incomplete sentence (lines 206-206)
Our reply: We have modified as recommended.
Misspelled word (forms in line 231)
Our reply: We have modified as recommended.

Round 2

Reviewer 2 Report

Clarification that Ype is a household detergent should be added to text, perhaps on line 86. “Household dishwashing detergent Ype Clear . . .”

I am still confused about the experimental design. Lines 118-128 talk about groups, but apparently “group” means a single petri dish. To avoid confusion, perhaps the phrases “petri dish” and “petri dishes” should be used instead of “group” and “groups” here. Lines 118 and 119 have been modified to include the G group, but then the G group is introduced separately as one of three additional groups in lines 128 and 129. It might make sense to refer to the inoculated eggs as groups, but a single petri dish is not a group.

Then Table 1 lists number of replicates for each “Group.” Are these replicates actually number of embryos inoculated? If so, these are not experimental replicates, and the column should be labeled “Number of eggs inoculated.” Actually this column is not necessary, since the information has been added that 4-6 eggs were inoculated and the number of eggs is given in Table 2. The first column might be labeled “Treatment” instead of “Group”

The authors seem to have misunderstood my statement about adjusting the results the viral RNA copies in the inoculum. If the results are presented as the number of viral RNA copies present in the virus suspension applied to the plates, then I expect the number of viral copies to be higher than the EID50 rather than 3000 times lower. I think they reported the number of viral RNA copies in the amount used in the RT-PCR reaction, which represents only a small proportion of the virus suspension used on the plate. In fact, the number of viral RNA copies in the virus applied to the plate is not relevant. There is no reason to compare them to the viral RNA copies recovered from the inoculated eggs (which have undergone an unknown number of replication cycles.  Therefore the results in lines 179-184 can be deleted. For the experiment, EID50 in the inoculum is important, but number of viral RNA copies is not. A reason for determining viral RNA copy number in the samples loaded onto the plates is given in the discussion (lines 237-238), but the results were not presented in a way that this purpose could be fulfilled. That the EID50 in the virus suspension applied to the plates was the same is more important and this was not tested. One should be able to make the assumption that if the same volume of the same suspension was applied to both film and control plates, that the same EID50 was applied. The sentence in lines 237-238 should also be deleted.

Figure 1: What do the error bars represent? SD or SEM? The format of the error bars is not the same for the two formulations. For the inanimate surface formulation there are no horizontal lines at the top of the error bars. As for Table 1, “hands” should be called “hands formulation: because it was not tested on hands.

I have noted some more language corrections:

Line 42: It should be “700 thousand deaths” and August should be capitalized

Line 57: It should be “on inanimate” instead of “in inanimate”

Line 70: It should be “another” instead of “other”

Line 75: genus should be singular (genus, not genera)

Line 82: It should be “applying to” instead of “applying in.”

Line 87:  It should be “other” instead of “others”

Lines 95 and 118: It should be “in diameter” instead of “or diameter.” Alternatively, it could say simply “85 mm diameter”

Line 103-104: It should be “film components” instead of “films components”

Line 121: It should be “at room temperature” instead of “in room temperature”

Lines 136 and 152: It should be “until processing” or “until being processed” instead of “until be processed”

Line 140: It should be “into eggs” instead of “in eggs” Also in line 180 it should be “into” instead of “in.”

Line 147: It should be “276 bp in size” instead of “with 276 bp of size.”  However, since the sentence refers to the target of the RT-PCR (not the product), the units should be nucleotides instead of base pairs, because the viral genome is single-stranded.

Line 222: It should be “on a plastic surface” instead of “in plastic surface”

Line 224: It should be “on” instead of “in.”

Line 227: It should be “for screening” or “to screen” instead of “to screening.”

Line 228: It should be “despite that this study” instead of “despite this study.” Alternatively it could say “despite this study having used ACoV as a virus model . . .”

Line 257: It should be “minimizing instead of minimize.”

Lines 266 and 269 and 286: It should be “on” instead of “in.”

Line 270: It should be “prevent SARS-CoV-2” instead of “prevent the SARS-CoV-2”

Line 275: It should be “through” instead of “thorough”

Line 276: It should be “hazardous chemicals” instead of “hazard chemical.”

Line 278: It should be “contaminate ecosystems” instead of “contaminates the ecosystems”

Line 279: It should be “previous ones, which” instead of “previously ones, that”

Line 280: It should be “environmental” instead of “environment”

Line 280: It should be “Hand lotion” instead of “The hand lotion.”

Line 282: It should be “washing, which is a common” instead of “washing that is common”

Line 289: It should be “mitigate” instead of “mitigates”

Author Response

Our reply: We would like to thank once more the reviewer for his/her thoroughness and expertise.  We appreciated also the time dedicated to do the critical review of our manuscript and, especially, for his (/her) constructive suggestions. We have made several modifications in the manuscript according with recommended by the reviewers. Now we will present our considerations to the specific comments of reviewer-2:

Clarification that Ype is a household detergent should be added to text, perhaps on line 86. “Household dishwashing detergent Ype Clear . . .

Our reply: We have modified as recommended (P2-L86)

I am still confused about the experimental design. Lines 118-128 talk about groups, but apparently “group” means a single petri dish. To avoid confusion, perhaps the phrases “petri dish” and “petri dishes” should be used instead of “group” and “groups” here. Lines 118 and 119 have been modified to include the G group, but then the G group is introduced separately as one of three additional groups in lines 128 and 129. It might make sense to refer to the inoculated eggs as groups, but a single petri dish is not a group.

Our reply: We are in agreement with the reviewer statement. We have modified the text replacing the term “group” by “petri dish” or control (especifically for G to I). (P2L119-130)

Then Table 1 lists number of replicates for each “Group.” Are these replicates actually number of embryos inoculated? If so, these are not experimental replicates, and the column should be labeled “Number of eggs inoculated.” Actually this column is not necessary, since the information has been added that 4-6 eggs were inoculated and the number of eggs is given in Table 2. The first column might be labeled “Treatment” instead of “Group”

Our reply: Yes, the column “replicates” was referred to the number of embryos inoculated. As recommended by the reviewer, we have removed this column. We have also replaced the term “group” by “treatment”.

The authors seem to have misunderstood my statement about adjusting the results the viral RNA copies in the inoculum. If the results are presented as the number of viral RNA copies present in the virus suspension applied to the plates, then I expect the number of viral copies to be higher than the EID50 rather than 3000 times lower. I think they reported the number of viral RNA copies in the amount used in the RT-PCR reaction, which represents only a small proportion of the virus suspension used on the plate. In fact, the number of viral RNA copies in the virus applied to the plate is not relevant. There is no reason to compare them to the viral RNA copies recovered from the inoculated eggs (which have undergone an unknown number of replication cycles.  Therefore the results in lines 179-184 can be deleted. For the experiment, EID50 in the inoculum is important, but number of viral RNA copies is not. A reason for determining viral RNA copy number in the samples loaded onto the plates is given in the discussion (lines 237-238), but the results were not presented in a way that this purpose could be fulfilled. That the EID50 in the virus suspension applied to the plates was the same is more important and this was not tested. One should be able to make the assumption that if the same volume of the same suspension was applied to both film and control plates, that the same EID50 was applied. The sentence in lines 237-238 should also be deleted.

Our reply: We are in agreement with the reviewer’s statement. We would like to apologize for the misunderstood. In the previous version of our manuscript, we have inserted the RNA virus quantification in the inoculum was inoculated in the study aiming to demonstrate that same RNA loads were present in both petri dish treated with film + virus and petri dish treated with just the virus (respective positive control group). However, as suggested by the reviewer we have removed the related results (including the sentence in the discussion  section).

Figure 1: What do the error bars represent? SD or SEM? The format of the error bars is not the same for the two formulations. For the inanimate surface formulation there are no horizontal lines at the top of the error bars. As for Table 1, “hands” should be called “hands formulation: because it was not tested on hands.

Our reply: The errors bars in Figure 1 are standard deviation observed for each treatment. We have corrected the format of error bars of inanimate surface treatments. We also have modified the term “Hands” to “Hands formulation” as suggested by the reviewer.

I have noted some more language corrections:

Line 42: It should be “700 thousand deaths” and August should be capitalized

Our reply: We have modified as recommended

Line 57: It should be “on inanimate” instead of “in inanimate”

Our reply: We have modified as recommended

Line 70: It should be “another” instead of “other”

Our reply: We have modified as recommended

Line 75: genus should be singular (genus, not genera)

Our reply: We have modified as recommended

Line 82: It should be “applying to” instead of “applying in.”

Our reply: We have modified as recommended

Line 87:  It should be “other” instead of “others”

Our reply: We have modified as recommended

Lines 95 and 118: It should be “in diameter” instead of “or diameter.” Alternatively, it could say simply “85 mm diameter”

Our reply: We have modified as recommended

Line 103-104: It should be “film components” instead of “films components”

Our reply: We have modified as recommended

Line 121: It should be “at room temperature” instead of “in room temperature”

Our reply: We have modified as recommended

Lines 136 and 152: It should be “until processing” or “until being processed” instead of “until be processed”

Our reply: We have modified as recommended.

Line 140: It should be “into eggs” instead of “in eggs” Also in line 180 it should be “into” instead of “in.”

Our reply: As we have removed the RNA virus quantification in the inoculum, the sentences containing these suggested modifications were removed.

Line 147: It should be “276 bp in size” instead of “with 276 bp of size.”  However, since the sentence refers to the target of the RT-PCR (not the product), the units should be nucleotides instead of base pairs, because the viral genome is single-stranded.

Our reply: We have modified as recommended. The term “276 bp of size”, was replaced by “276 nucleotides in size” (P4L168).

Line 222: It should be “on a plastic surface” instead of “in plastic surface”

Our reply: We have modified as recommended.

Line 224: It should be “on” instead of “in.”

Our reply: We have modified as recommended.

Line 227: It should be “for screening” or “to screen” instead of “to screening.”

Our reply: We have modified as recommended.

Line 228: It should be “despite that this study” instead of “despite this study.” Alternatively it could say “despite this study having used ACoV as a virus model . . .”

Our reply: We have modified as recommended.

Line 257: It should be “minimizing instead of minimize.”

Our reply: We have modified as recommended.

Lines 266 and 269 and 286: It should be “on” instead of “in.”

Our reply: We have modified as recommended.

Line 270: It should be “prevent SARS-CoV-2” instead of “prevent the SARS-CoV-2”

Our reply: We have modified as recommended.

Line 275: It should be “through” instead of “thorough”

Our reply: We have modified as recommended.

Line 276: It should be “hazardous chemicals” instead of “hazard chemical.”

Our reply: We have modified as recommended.

Line 278: It should be “contaminate ecosystems” instead of “contaminates the ecosystems”

Our reply: We have modified as recommended.

Line 279: It should be “previous ones, which” instead of “previously ones, that”

Our reply: We have modified as recommended.

Line 280: It should be “environmental” instead of “environment”

Our reply: We have modified as recommended.

Line 280: It should be “Hand lotion” instead of “The hand lotion.”

Our reply: We have modified as recommended.

Line 282: It should be “washing, which is a common” instead of “washing that is common”

Our reply: We have modified as recommended.

Line 289: It should be “mitigate” instead of “mitigates”

Our reply: We have modified as recommended.
